# Self-Supervised Learning of Motion Concepts by Optimizing Counterfactuals

**Stefan Stojanov**[*]
Stanford University

**David Wendt**[*]
Stanford University

**Seungwoo Kim**[*]
Stanford University

**Rahul Mysore Venkatesh**[*]
Stanford University

**Kevin Feigelis**
Stanford University

**Klemen Kotar**
Stanford University

**Khai Loong Aw**
Stanford University

**Jiajun Wu**
Stanford University

**Daniel L.K. Yamins**
Stanford University

## Abstract

Estimating motion primitives from video (e.g., optical flow and occlusion) is a critically important computer vision problem with many downstream applications, including controllable video generation and robotics. Current solutions are primarily supervised on synthetic data or require tuning of situation-specific heuristics, which inherently limits these models' capabilities in real-world contexts. A natural solution to transcend these limitations would be to deploy large-scale, self-supervised video models, which can be trained scalably on unrestricted real-world video datasets. However, despite recent progress, motion-primitive extraction from large pretrained video models remains relatively underexplored. In this work, we describe Opt-CWM, a self-supervised flow and occlusion estimation technique from a pretrained video prediction model. Opt-CWM uses "counterfactual probes" to extract motion information from a base video model in a zero-shot fashion. The key problem we solve is optimizing the quality of these probes, using a combination of an efficient parameterization of the space counterfactual probes, together with a novel generic sparse-prediction principle for learning the probe-generation parameters in a self-supervised fashion. Opt-CWM achieves state-of-the-art performance for motion estimation on real-world videos while requiring no labeled data. [1]

## 1 Introduction

Extracting "low-level" scene motion properties such as optical flow [13, 40], occlusions [28], and point or object tracks [19, 10] is important for video understanding applications such as automated video filtering [54, 55], action recognition [26, 38] and motion prediction [5, 53]. Recently, scene motion primitives have also been critical for increasing the controllability and consistency of video generation models [15], and have gained an important role in robotics applications [43, 4].

Optical flow and occlusion are two core primitives in this domain. The most common approach to optical flow estimation uses supervised learning from labeled flow data. However, because densely annotating flow in real-world scenes is prohibitively expensive, supervised methods usually rely on synthetic data [31, 32]. Methods trained on synthetic data have proven to be robust in real-world video [40, 50]. However, relying on this approach has limited flow estimation methods from taking

---

[*]Equal contribution.
[1]Project website: `https://neuroailab.github.io/opt_cwm_page/`

39th Conference on Neural Information Processing Systems (NeurIPS 2025).

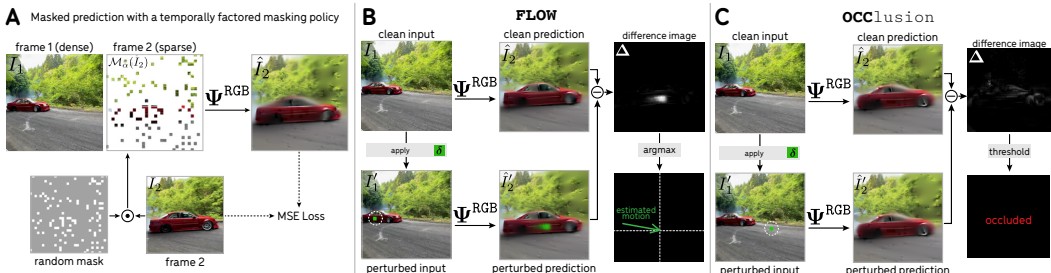

Figure 1: **Counterfactual probing for flow and occlusion:** **(A)** CWMs learn to predict the next frame with a temporally factored masking policy [3]. **(B)** The motion of a point can be estimated using a counterfactual probing program FLOW: the model predicts the next frame with and without a perturbation placed on the point, and the difference image between the clean and perturbed predictions reveals the estimated motion. **(C)** Occlusion is estimated using a related probe OCC: a diffuse and low magnitude difference indicates that the perturbed point has been occluded.

advantage of recent advances in self-supervised visual representation learning from massive video datasets [41, 35, 14] and inherently has to contend with a sim-to-real domain gap.

In contrast, self-supervised motion-estimation methods are typically based on *photometric loss* – learning frame-pair feature correspondences to warp pixels from one RGB frame to corresponding locations in future frames. However, pure photometric loss is a weak constraint, in part because correspondences are often ill-defined (e.g., objects with homogeneous textures). Existing state-of-the-art methods use various nearest neighbor or clustering procedures [21, 6], or complement photometric loss with strong task-specific regularizations like smoothness [23, 39]. Because these heuristics are often only correct in narrow scenarios, performance is limited in cases where the heuristics fail.

In this work, we show how to extract high-quality self-supervised flow and occlusion estimates without the use of such heuristics. A promising initial approach to this problem comes from Counterfactual World Modeling (CWM) [3, 44], a method that constructs zero-shot estimates of a variety of visual properties (flow, segments, shape, etc) from an underlying pre-trained multi-frame model (Figure 1). CWM begins with a *sparse RGB-conditioned next frame predictor* $\Psi^{RGB}$, a two-frame masked autoencoder trained with a highly asymmetric masking policy [3]. This forces the model to encode scene dynamics in a small number of patch feature tokens that *factor* temporal dynamics from visual appearance. Motion properties can then be extracted from the base model in a zero-shot fashion via simple "counterfactual probes", acting as a kind of test-time inference procedure. For example, to compute flow from a given point in the first frame, a perturbation is made to the image at that point, and flow is computed by comparing the difference between $\Psi^{RGB}$'s prediction on the perturbed (counterfactual) condition with its prediction in the original unperturbed (factual) condition (see Figure 1B). Intuitively, this corresponds to placing a visual marker on the point, predictively flowing it forward, and then analyzing where it gets "carried" in the predicted next frame.

In principle, the CWM approach circumvents the key limitation of the heuristic-based methods by replacing situation-specific fixed heuristics (e.g., motion smoothness) with a general-purpose predictive model. The quantity of interest, in our case flow, is defined as the outcome of probing the model's predictions [44]. However, while CWM is an intriguing conceptual proposal, it has a conceptual drawback that substantially limits its real-world performance: the probes that it relies on are hand-designed and can be out-of-domain in real-world video. Perturbations are often not properly "carried along" with moving objects, resulting in suboptimal counterfactual motion extractions (Figure 2B). As a result, the accuracy of the flows extracted by the originally proposed CWM method has remained inferior to state-of-the-art flow estimation solutions.

Here we present Opt-CWM, a generic solution to this problem. Opt-CWM introduces two conceptual innovations that leverage the advantages of the CWM idea while making it highly performant in real-world settings. The first of these innovations is a method for *parameterizing* a counterfactual probe policy generator with a learnable neural network (Figure 2A). This network can predict situation-specific probes that take into account the appearance context (both local and global) around target points to be tracked, and thus can be less out-of-distribution than hand-coded probes. The second innovation is an approach for *learning* the probe generator in a principled fashion without relying

on any supervision from labeled data or heuristics. The main insight behind this learning procedure is to construct a task-agnostic generalization of the asymmetric masking principle used to train the base model $\Psi^{\texttt{RGB}}$ itself. In particular, Opt-CWM connects sparse outputs of the parameterized flow prediction function to a randomly initialized sparse flow-conditioned next-frame predictor $\Psi^{\texttt{flow}}$ and performs joint optimization (Figure 3) of both $\Psi^{\texttt{flow}}$ and the probe generator. This forces $\Psi^{\texttt{flow}}$ to predict a future frame based on a present frame and sparse (putative) flow, creating an information bottleneck that generates useful gradients back on the probe generator's parameters.

We find that Opt-CWM achieves strong performance when compared with existing motion estimation methods (both supervised and self-supervised) that are purposely built for this task [39, 37, 50, 33], as well as recent adaptations of large-scale self-supervised visual representations for motion estimation [23], when evaluated on real-world benchmarks [10]. The success of our approach reveals a promising direction for scalable counterfactual extraction of a variety of visual properties.

## 2   Related Work

**Supervised flow estimation.** Supervised methods like RAFT [40, 50] approach optical flow as a dense regression problem and learn from synthetic optical flow datasets [7, 31]. They also typically use task-specific architectures that are tailor-made for optical flow estimation, with strong inductive biases (e.g., iterative flood-filling) and task-specific regularizations to ensure learning from limited training datasets. While these methods show strong performance in many contexts, their reliance on synthetic supervision and specialized architectures limits their generalizability. It is for this reason that our self-supervised Opt-CWM, which can be trained on unlimited in-the-wild videos, can outperform even supervised methods in certain key contexts.

**Self-supervised flow with photometric loss.** Methods for self-supervised flow learning [24, 39, 27], such as SMURF [39], learn dense visual correspondence by optimizing photometric loss. Because of the weakness of pure photometric loss alone as supervision, these methods rely on a complex variety of heuristically chosen regularization losses (e.g., spatial smoothness of flow, among others) to achieve reasonable performance levels. Because these heuristics need to be tuned in a dataset-specific manner, these methods have failure models in complex dynamic scenes, especially with variable and large time-frame gaps. In contrast to these methods, Opt-CWM does not rely on such heuristics, as the quality of the flow extraction is directly correlated with the prediction learning objective.

**Augmenting self-supervised flow with visual pre-training.** A variety of methods augment photometric loss using features derived from self-supervised visual pre-training [2, 6, 51, 23]. For example, the recent state-of-the-art Doduo method [23] uses DINOv2 features as a basis on which to compute feature correspondences for downstream photometric loss. This approach allows the extension of these methods to wider video training datasets (such as Kinetics) and thereby improves performance and generalizability. However, even when backed by strong image features, photometric loss is a weak constraint, requiring additional heuristic regularizers to improve performance. Opt-CWM, by avoiding scenario-specific heuristics, compares favorably to these methods.

**Point tracking.** Point tracking across multiple frames is a related problem to flow and occlusion estimation. The majority of solutions for point tracking are supervised [19, 10] or semi-supervised [25, 11], and as such are further out of scope for this work. However, several recent works propose self-supervised approaches to finding temporal correspondence, typically relying on pre-trained representations [6, 21]. These methods then extract point tracks through consistency objectives such as cycle consistency [6, 21, 37] or heuristics like softmax-similarity [45] applied at the frame pair level. Tumanyan et al. [42] take a related approach, performing test-time optimization on individual videos using pre-trained DINO features and short-term supervision from RAFT. The current state-of-the-art self-supervised method in this domain, GMRW [37], which is the main baseline comparison for our proposed Opt-CWM, uses cycle consistency to build tracks based on frame pair-level predictions.

**Real-world motion benchmarks.** The TAP-Vid benchmark [10] provides a critical set of metrics for measuring the accuracy of motion-estimation systems in real-world video. This is critical for ensuring that potential advances in motion estimation are tested against the challenges of real-world motion complexities, covering scenarios not encountered in synthetic benchmarks (e.g., non-rigid, highly articulated, deformable and breakable objects, fluids, inelastic collisions, animate objects, and human interactions). While originally intended for the supervised point tracking domain, recent

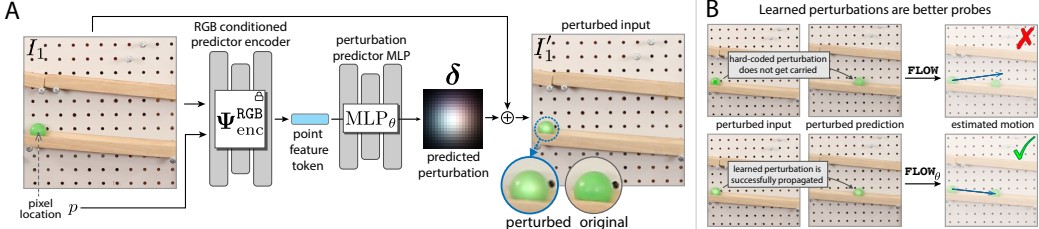

Figure 2: **Parameterizing the counterfactual probe generator as an input-conditioned function.**
(**A**) Building on a pre-trained RGB-conditioned predictor $\mathbf{\Psi}^{\text{RGB}}$, Opt-CWM uses an image-conditioned perturbation prediction function $\delta_\theta$ containing a small $\text{MLP}_\theta$. As illustrated in **B**, $\delta_\theta$ can learn to predict image-conditioned perturbations that blend naturally with the underlying scene, potentially allowing for the perturbation to be accurately carried over to the next frame prediction. But how should the parameters of $\delta_\theta$ be learned without any flow supervision labels? See Figure 3.

self-supervised tracking works have begun to utilize TAP-Vid as a main benchmark for motion estimation [23, 37]. In this work, we also follow this practice.

# 3 Methods

## 3.1 Counterfactual World Modeling

**RGB-Conditioned Next Frame Predictor.** A Countefactual World Model (CWM) is an RGB-conditioned next frame predictor $\mathbf{\Psi}^{\text{RGB}}$, consisting of an encoder $\mathbf{\Psi}^{\text{RGB}}_{\text{enc}}$ and decoder $\mathbf{\Psi}^{\text{RGB}}_{\text{dec}}$, similar to a VideoMAE [41], but trained with a highly asymmetric masking policy that reveals all patches of the first frame and a small fraction of patches of the second frame [3] (and see Figure 1A). Specifically, let $I_1, I_2 \in \mathbb{R}^{3 \times H \times W}$ be two frame pairs in a video, and define $\mathcal{M}_\alpha$ as a masking function that randomly masks some fraction, $\alpha$, of patches in an image. Given a fully visible first frame $I_1$ and a partially visible second frame $\mathcal{M}_\alpha(I_2)$, $\mathbf{\Psi}^{\text{RGB}}$ is trained to predict $I_2$ by minimizing

$$\mathcal{L} = \text{MSE}(\hat{I}_2, I_2), \text{ where } \hat{I}_2 = \mathbf{\Psi}^{\text{RGB}}\big(I_1, \mathcal{M}_\alpha(I_2)\big). \tag{1}$$

Here we train CWM with $\alpha = 0.1$ on publicly available video data with a frame gap of 150ms. (See the supplement for more details.) The asymmetric masking training policy forces $\mathbf{\Psi}^{\text{RGB}}$ to separate scene appearance—which is wholly available in the first frame—from scene dynamics, the information of which is now concentrated in the sparse set of visible next frame patches. In other words, $\mathbf{\Psi}^{\text{RGB}}$ is "temporally factored".

**Motion Estimation With Counterfactual Probes.** Because it induces strong dependence on the appearance and position of the revealed patches from $I_1$ and $I_2$, temporal factoring allows the zero-shot extraction of visual structure through applying counterfactual probes: small changes to the appearance or the position of visible patches. By measuring the predictor's response to these counterfactuals, we can easily extract useful information like object motion, segments, or shape from its representation [3]. As shown in Figure 1B, using the FLOW procedure, a colored patch is placed on a moving object, and its motion can be determined by finding its location in the predicted frame. To track some pixel location $p_1 = (\text{row}_1, \text{col}_1)$ from one frame to the next, input image $I_1$ is perturbed by adding a colored patch $\delta$ at pixel location $p_1$ to create the counterfactual input image $I'_1 = I_1 + \delta$. Then, the next frames with and without the counterfactual perturbation are predicted:

$$\hat{I}'_2 = \mathbf{\Psi}^{\text{RGB}}\big(I_1 + \delta, \mathcal{M}_\alpha(I_2)\big) = \mathbf{\Psi}^{\text{RGB}}\big(I'_1, \mathcal{M}_\alpha(I_2)\big), \text{ and } \hat{I}_2 = \mathbf{\Psi}^{\text{RGB}}\big(I_1, \mathcal{M}_\alpha(I_2)\big). \tag{2}$$

Subtracting these two predicted frames and taking an $L_1$-norm across the color channels produces the difference image $\mathbf{\Delta} = |\hat{I}'_2 - \hat{I}_2|^c_1$. Finally, the next-frame pixel location $\hat{p}_2$ can be computed by finding the peak in the difference image: $\hat{p}_2 = \arg\max \mathbf{\Delta}$. FLOW is essentially a kind of test-time inference applied to the pretrained CWM base model. To extract occlusion information, the OCC procedure is identical to FLOW up to computation of the difference image $\mathbf{\Delta}$ (See Figure 1). However, if a patch in the first frame gets occluded in the second frame, the response to the perturbation in the difference image $\mathbf{\Delta}$ will be small in magnitude and diffuse in shape. Applying a simple threshold to the maximum value of $\mathbf{\Delta}$ creates an occlusion binary indicator.

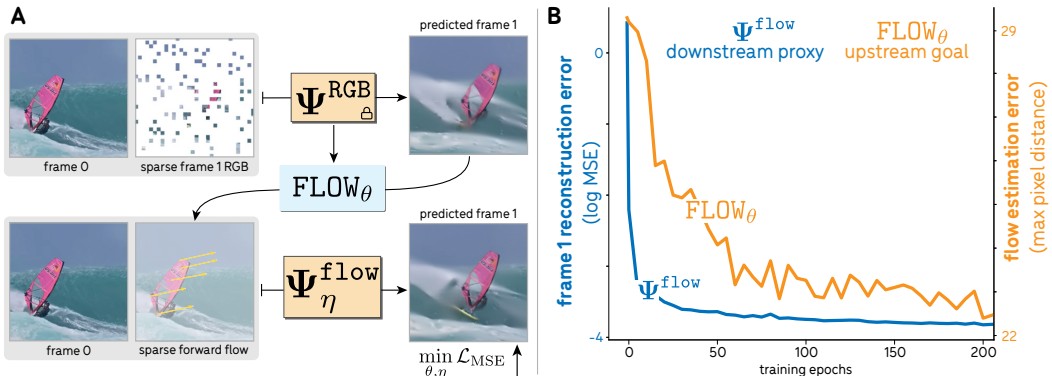

Figure 3: **A generic sparse-prediction principle for learning optimized counterfactuals. A)** The parameterized counterfactual flow function FLOW$_\theta$ extracts motion from a frozen RGB-conditioned predictor $\mathbf{\Psi}^{\text{RGB}}$ through counterfactual perturbation (details in Figure 2). Its parameters $\theta$ are trained using gradients from a flow-conditioned predictor $\mathbf{\Psi}^{\text{flow}}_\eta$ that is jointly trained to perform next-frame prediction. The predictor $\mathbf{\Psi}^{\text{flow}}$ *can only learn to predict future frames if it is given correct flow-like vectors*, a form of information bottleneck that ensures useful gradients are passed back to FLOW$_\theta$. We thus exploit the pre-trained $\mathbf{\Psi}^{\text{RGB}}$ predictor by bootstrapping a flow-conditioned predictor $\mathbf{\Psi}^{\text{flow}}$, using an extension of the principle of sparse next-frame prediction. **(B)** There is tight coupling between the flow-conditioned predictor $\mathbf{\Psi}^{\text{flow}}$ and the learned flow estimation function FLOW$_\theta$, so both pixel reconstruction (the proxy goal) and motion estimation (our real goal) simultaneously improve.

## 3.2 Optimizing Counterfactual Perturbations

**The problem with hand-designed perturbations.** While the CWM approach of using fixed hand-designed probes (e.g., additive colored squares) can sometimes be effective in probing motion with $\mathbf{\Psi}^{\text{RGB}}$, they are often suboptimal. First, this is because they are *out of domain* for the base predictor, and second, by being image- and position-independent, they can be unsuited to the local image context. Anecdotally, this results in visually obvious failure cases, such as the perturbation not moving with the object or being suppressed entirely.

Using the challenging TAP-Vid benchmark (see Section 4 for more details), we empirically quantified that the original fixed hand-designed perturbations are insufficient for self-supervised motion estimation performance (see CWM results in Table 1). The main requirement for a "good" perturbation is that it is sufficiently in-distribution and image/point specific to cause meaningful context-dependent changes for probing the base predictor. But how can probes be designed for this purpose? Our solution has two basic novel components: *parameterizing* an image-conditioned and differentiable counterfactual probe generator, and formulating a general-purpose self-supervised loss objective for *learning* the probe generation policy parameters.

**Parameterized Perturbations.** We re-formulate the motion extraction procedure from Section 3.1 to make it a parameterized differentiable policy function and introduce the functional form of a sum of colored Gaussians as a natural perturbation class. (See Figure 2A)

Let FLOW$_\theta$: $(I_1, I_2, p_1) \mapsto \hat{\varphi}$ be a per-pixel motion estimation function with learnable parameters $\theta$ that takes an image pair $(I_1, I_2)$ and a pixel location $p_1$ in $I_1$ and outputs the predicted flow $\hat{\varphi} = \hat{p}_2 - p_1$. Here, $\hat{p}_2$ is the estimated second frame pixel location. The procedure FLOW$_\theta$ involves multiple components: the counterfactual perturbation function, $\delta_\theta(I_1, \mathcal{M}_\alpha(I_2), p_1)$, which now produces variable counterfactual perturbations as a function of the frame pair and pixel location (as opposed to a fixed perturbation $\delta$, used in the standard CWM); the pre-trained, frozen, RGB-conditioned predictor, $\mathbf{\Psi}^{\text{RGB}}$, as utilized within the FLOW$_\theta$ program; and a "softargmax" module to predict a pixel location using a differentiable approximation to the argmax function.

*Gaussian Perturbations.* We choose to parameterize the counterfactual perturbations as Gaussians because this function class presents a natural method of forming in-domain counterfactual input images. To compute the Gaussian parameters for a given counterfactual perturbation, we use the

encoder of the RGB-conditioned predictor, $\mathbf{\Psi}_{\text{enc}}^{\text{RGB}}$. This outputs a sequence of feature tokens from its last transformer block, which encode global and local scene content for each patch and thus form a natural basis from which Gaussian parameters can be computed using a shallow MLP. Given a pixel location $p_1$, we find its corresponding patch embedding token, $\mathbf{t}_{p_1}$, and use it as an input to an MLP that outputs a parameter vector which is in turn used to compute the Gaussian perturbation:

$$\delta_\theta(I_1, \mathcal{M}_\alpha(I_2), p_1) = \text{Gaussian}\left(\text{MLP}_\theta\left(\mathbf{t}_{p_1}\right)\right) \text{ where } \mathbf{t}_{p_1} = \mathbf{\Psi}_{\text{enc}}^{\text{RGB}}(I_1, \mathcal{M}_\alpha(I_2))_{p_1}. \tag{3}$$

Then, $\texttt{FLOW}_\theta$ computes the difference image, $\mathbf{\Delta}$, similar to the $\texttt{FLOW}$ program, using $\hat{I}_2' = \mathbf{\Psi}^{\text{RGB}}\left(I_1 + \delta_\theta, \mathcal{M}_\alpha(I_2)\right)$. To make $\texttt{FLOW}_\theta$ differentiable, we use a softargmax over $\mathbf{\Delta}$ to estimate $\hat{p}_2$.

*Softargmax Module.* We follow the softargmax formulation proposed in [48]. Given a difference image, $\mathbf{\Delta} = |\hat{I}_2' - \hat{I}_2|_1^c$, we apply a temperature-scaled 2D softmax and then take the expectation to find $\hat{p}_2 = \mathbb{E}_{p_2 \sim \text{softmax}(\mathbf{\Delta}/\tau)}[p_2]$. The predicted flow is then computed as $\hat{\varphi} = \hat{p}_2 - p_1$.

**Learning Optimized Counterfactuals.** Now that the perturbation generator is parameterized, the question arises: how can its parameters be learned? What type of self-supervised objective will cause the perturbation generator function to be context-specific and result in accurate flow vectors? Our main insight is that this problem can be "bootstrapped" in a robust fashion by generalizing the sparse asymmetric mask learning paradigm to encompass a coupled and mixed-mode RGB-Flow prediction problem without using labeled data (see Figure 3). Specifically, we jointly train the parameterized counterfactual motion prediction function, $\texttt{FLOW}_\theta$, which estimates a set of flow vectors; together with a sparse flow-conditioned predictor, $\mathbf{\Psi}^{\texttt{flow}}$, which takes a single frame along with sparse flow vectors to predict the next frame. We constrain $\texttt{FLOW}_\theta$ by passing its outputs as inputs to $\mathbf{\Psi}^{\texttt{flow}}$ and training end-to-end using final RGB reconstruction loss on the predictions of $\mathbf{\Psi}^{\texttt{flow}}$. As $\mathbf{\Psi}^{\texttt{flow}}$ has no access to any RGB patches from the second frame $I_2$, it is only if the putative flows are correct that it be possible for $\mathbf{\Psi}^{\texttt{flow}}$ to use them to minimize the next-frame reconstruction loss.

Specifically, given an image pair $(I_1, I_2)$, we estimate the motion for a set of pixels $\mathcal{P} = \{p_1^{(1)}, p_1^{(2)}, \ldots, p_1^{(n)}\}$ using $\texttt{FLOW}_\theta$, obtaining a set of estimated forward flow vectors $\hat{\mathcal{F}} = \{\hat{\varphi}^{(1)}, \hat{\varphi}^{(2)}, \ldots, \hat{\varphi}^{(n)}\}$. Let $\mathbf{\Psi}_\eta^{\texttt{flow}} \colon (I_1, \hat{\mathcal{F}}) \mapsto \hat{I}_2$ be a flow-conditioned next frame predictor with parameters $\eta$ that takes the first frame RGB input $I_1$ and predicts the next frame $\hat{I}_2$, conditioned on the flow input $\hat{\mathcal{F}}$. We jointly optimize $\theta$ and $\eta$, by minimizing $\min_{\theta,\eta} \mathcal{L}_{\text{MSE}}(\hat{I}_2, I_2)$. Figure 3B shows that optimizing end-to-end reconstruction does indeed couple tightly to upstream flow accuracy, as required for effective bootstrapping.

In this work, we investigate two $\mathbf{\Psi}^{\text{RGB}}$ base predictors, with 175M and 1B learnable parameters. For optimizing the counterfactuals and ablations, we use the 175M model, and report benchmark results by applying the learned counterfactual probes to the 1B model.

**Inference-time Enhancements.** A simple random masking strategy may inadvertently reveal the ground truth RGB at the next frame location we are trying to predict for a particular point. In this event, the model will not carry over the counterfactual perturbation to the future frame, leading to an erroneous flow prediction. A simple yet effective inference-time solution is *multi-mask* (MM), in which we apply multiple random masks and average across the resulting delta images to reduce the influence of sub-optimal masks. Following prior work [13, 22], we also perform an iterative multiscale refinement of flow predictions by recursively applying $\texttt{FLOW}_\theta$ to smaller crops centered on the predicted point location, $\hat{p}_2$ of the previous iteration. We observe that $\texttt{FLOW}_\theta$ is able to generate good initial flow predictions, and thus benefits from refinement (Table 2).

## 4  Experiments

**Evaluation Protocol.** Our main datasets for evaluation are TAP-Vid DAVIS and TAP-Vid Kinetics [10], the DAVIS [34] and Kinetics [26] datasets with human flow and occlusion annotations, along with the synthetic Kubric [18] dataset where ground-truth flows and occlusions are known. All the methods we test output direct two-frame flow (point-to-point correspondence) predictions. Some of them output occlusion predictions, which we use when available. For flow methods without an existing implementation of occlusion prediction, we use cycle consistency to compute occlusion estimates: occlusion is the event of inconsistency between forward and backward predictions greater than 6 pixels. Models that can accept variable resolution inputs are run with the resolution closest to native that can be fit into memory, ensuring that each is run optimally. Metrics for both procedures

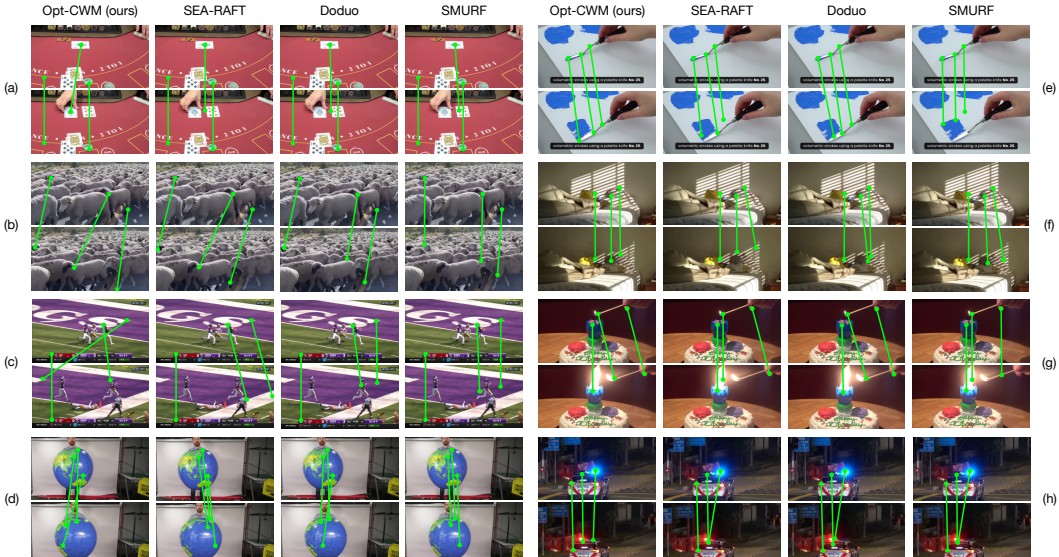

Figure 4: **Qualitative comparisons on real-world videos.** These examples show the failure modes of methods reliant on visual similarity. We observe that the baseline models struggle against subtle but functionally important changes in largely homogeneous scenes depicting objects of similar color and texture ((a) - (e)). Further, the use of photometric loss in self-supervised methods such as SMURF can also be susceptible to differences in light intensity across frame pairs ((f) - (h)). In contrast, as a visual world model, Opt-CWM possesses a holistic understanding of scene transformations and object dynamics, and is able to find correspondence without arbitrary heuristics.

are always computed after rescaling predictions to $256 \times 256$ resolution, as in [37]. Following the TAP-Vid First protocol proposed in [10], for each point, we take the frame in which it is first visible and track its motion only forward in time. We also use the first frame a point is visible as the reference frame, and track points in future time steps with reference to this frame. This is a challenging setting as it involves tracking points across variable and often large frame gaps. It is also most comparable to many real-world use-case scenarios where the frame gap may be unknown or uncontrollable. We show additional results for the TAP-Vid Constant five-Frame Gap (CFG) protocol in the supplementary material, which is more favorable to standard optical flow methods.

**Metrics.** We use the official metrics from the TAP-Vid evaluation protocol [10]: 1) *Average Jaccard* (AJ), a precision metric measuring a combination of point tracking and occlusion prediction; 2) *Average Distance* (AD) between the estimated pixel and ground truth locations; 3) $< \delta^x_{avg}$, which measures the average percentage of points predicted correctly within a variety of pixel distance thresholds; and 4) *Occlusion Accuracy* (OA), the fraction of points correctly predicted as either occluded or visible. Additionally, to account for the relative lack of occlusion events in the dataset, we also evaluate 5) *Occlusion F1* (OF1), which computes the F1 score of the occlusion predictions.

**Baselines.** Our evaluation protocol requires tracking points in videos through occlusion by finding temporal correspondence: given a frame pair, determine where the point went or whether it was occluded. Therefore, the appropriate baselines are supervised and self-supervised optical flow methods, and self-supervised temporal correspondence methods. We run the following baselines:

*CWM* [3, 44] represents motion estimated through counterfactual extractions with a fixed hand-designed perturbation. This comparison shows how the innovations introduced by Opt-CWM lead to very substantial performance improvements.

*GMRW* [37] is a self-supervised video correspondence approach that trains a transformer using cycle consistency on contrastive random walks. GMRW is designed for temporal correspondence-based long-range tracking and is the SOTA baseline for comparison on TAP-Vid First.

*SMURF* [39] is an unsupervised method specifically designed for optical flow estimation. SMURF tailors the RAFT [40] architecture so it can be trained using a combination of optical flow-specific

Table 1: **Quantitative results on TAP-Vid dataset.** In the TAP-Vid First protocol, a point is tracked from when it is first visible to the end of the video, requiring motion estimation across large frame gaps. Opt-CWM outperforms both supervised and unsupervised baselines. "S" and "U" indicate supervised and unsupervised, respectively. Doduo is not strictly unsupervised as it uses segmentation labels. GMRW is trained on the Kubric dataset, (marked with ‡), making it a more favorable evaluation setting for that method because of the minimal domain gap. Best performing supervised models (shaded) are considered separately.

| | Method | DAVIS | | | | | Kinetics | | | | | Kubric | | | | |
|---|---|---|---|---|---|---|---|---|---|---|---|---|---|---|---|---|
| | | AJ↑ | AD↓ | $<\delta^x_{avg}$↑ | OA↑ | OF1↑ | AJ↑ | AD↓ | $<\delta^x_{avg}$↑ | OA↑ | OF1↑ | AJ↑ | AD↓ | $<\delta^x_{avg}$↑ | OA↑ | OF1↑ |
| S | CoTracker-v3 [25] | 39.85 | 19.04 | 57.96 | 76.87 | 47.33 | 38.11 | 25.00 | 53.41 | 77.29 | 59.43 | 76.41 | 5.10 | 87.79 | 90.48 | 71.06 |
| | RAFT [40] | 41.77 | 25.33 | 54.37 | 66.40 | 56.12 | 41.92 | 23.00 | 53.49 | 74.67 | 70.62 | 71.93 | 5.60 | 82.15 | 88.54 | 68.34 |
| | SEA-RAFT [50] | 43.41 | 20.18 | 58.69 | 66.34 | 56.23 | 33.40 | 30.72 | 46.11 | 64.55 | 64.39 | 75.06 | 6.54 | 84.63 | 89.50 | 70.92 |
| | DPFlow [33] | 49.20 | 16.86 | 62.51 | 71.51 | 60.15 | 47.33 | 17.65 | 58.74 | 81.98 | 77.00 | 78.45 | 5.05 | 86.81 | 90.86 | 74.53 |
| U† | Doduo [23] | 23.34 | 13.41 | 48.50 | 47.91 | 49.43 | 31.51 | 15.05 | 46.87 | 66.71 | 66.01 | 54.98 | 5.31 | 72.20 | 73.56 | 52.67 |
| U | GMRW [37] | 36.47 | 20.26 | 54.59 | 76.36 | 42.85 | 25.58 | 29.28 | 41.63 | 71.05 | 33.57 | 58.36‡ | 3.84‡ | 79.27‡ | 80.70‡ | 32.18‡ |
| | SMURF [39] | 30.64 | 27.28 | 44.18 | 59.15 | 46.91 | 33.33 | 32.56 | 44.37 | 66.60 | 60.90 | 65.81 | 6.81 | 80.57 | 87.91 | 58.42 |
| | CWM [3, 44] | 15.00 | 23.53 | 26.30 | 76.63 | 18.22 | 17.60 | 26.43 | 29.61 | 72.59 | 28.95 | 28.77 | 11.64 | 41.63 | 84.93 | 11.35 |
| | Opt-CWM 1B (ours) | 51.88 | 7.70 | 68.63 | 80.44 | 68.43 | 47.03 | 11.25 | 61.31 | 80.74 | 76.21 | 79.98 | 3.36 | 89.40 | 90.11 | 72.56 |

heuristic losses like photometric loss and a variety of types of smoothness regularizations. SMURF is trained on synthetic datasets often used for optical flow estimation learning.

*Doduo* [23] applies a SMURF-like combination of self-supervised photometric and smoothness losses, scaling them to larger model architectures and in-the-wild training videos [52]. It leverages the DINOv2 [8] encoder to incorporate strong image priors. The Doduo model and training dataset are comparable in size to Opt-CWM, providing a control to ensure that the improved performance of Opt-CWM relative to SMURF is not solely due to model or dataset training size.

*SEA-RAFT* [50] is a supervised flow method building on RAFT [40] by adding additional pretraining on TartanAir [49], a novel mixture of Laplace loss, and improved initialization of the flow estimation.

*DPFlow* [33] is a very recent supervised method that leverages architectural advances to train against high-resolution flow data, leading to improvements relative to SEA-RAFT in many evaluations.

*CoTracker-v3* [25] is a supervised multi-frame model. We evaluate it as a two-frame model, since the multi-frame evaluation protocol is not comparable to Opt-CWM or the other baselines.

**Results.** We present our main results in Table 1. Opt-CWM outperforms all other self-supervised baselines for all datasets, as well as the supervised methods in most cases. In particular, Opt-CWM especially improves upon AD, demonstrating robustness even in difficult (though more realistic) cases with long frame gaps or high motion. The gap is especially large on real-world datasets such as DAVIS, where the baselines struggle with videos violating the heuristic assumptions for which they were optimized. Our experiments on the synthetic Kubric dataset [18], which is more favorable to methods trained on synthetic data, demonstrate that Opt-CWM has the best performance in this out-of-domain scenario.

Qualitatively, Opt-CWM makes strong use of its underlying world model, allowing it to accurately track a point's movement through long frame gaps and complex dynamics, including changes of lighting conditions. SEA-RAFT, Doduo, and SMURF, which lack an explicit dynamic world model, often lose track when the tracked object rotates, when lights turn on or off, or when shadows move (Figure 4). Further qualitative examples, including videos, can be found in the supplement.

**Ablations and Hyperparameter Analysis.** We perform several ablation studies of Opt-CWM. First (Table 2, left), we compare Opt-CWM with a spectrum of types of hard-coded perturbations, representing various forms of unoptimized CWM baseline, and find that learned interventions perform substantially better (see Table 2, left). The highly image-dependent nature of the optimized predicted perturbations is illustrated in the supplement. Increasing input resolution, the multi-mask inference (MM), and multiscale refinement (MS) procedures all improve performance.

We also study the effect of the core hyperparameters of our procedure both in training and inference (Table 3). We find that asymmetric masking during training is critical (which is likely why masked video models with standard masking procedures, such as VMAE [47], do not perform well at flow extraction), but that our model is highly stable to parameter choices at inference time.

Table 2: **(Left) Ablations.** We evaluate multi-mask (MM) and multiscale (MS), in addition to comparing our optimized perturbations ("learned") with the fixed ones ("red square"/"green square") [3, 44]. MM and MS columns indicate the number of masking or zooming iterations. We observe a clear improvement on all metrics, highlighting the need for bespoke, in-distribution counterfactual perturbations, multi-mask inference and multi-scale refinement. **(Right) Distillation into DPFlow.** For fast inference, we distill Opt-CWM into the small and efficient DPFlow architecture by sparsely pseudo-labeling Kinetics with Opt-CWM. This approach outpeforms the self-supervised SMURF and is competitive with the supervised models, while requiring no labeled training data.

| Type | MM | MS | Res. | AJ↑ | AD↓ | $< \delta_{avg}^x$↑ | OA↑ | OF1↑ |
|---|---|---|---|---|---|---|---|---|
| learned | 10 | 4 | 512 | **47.53** | **8.73** | **64.83** | 80.87 | **60.74** |
| learned | 1 | 4 | 512 | 42.85 | 9.82 | 59.72 | 78.55 | 60.20 |
| learned | 10 | 0 | 512 | 32.71 | 11.98 | 49.20 | 79.28 | 41.45 |
| learned | 3 | 2 | 512 | 40.51 | 9.72 | 58.57 | 80.34 | 50.06 |
| red square | 3 | 2 | 512 | 21.37 | 18.25 | 36.31 | 75.38 | 27.21 |
| green square | 3 | 2 | 512 | 30.44 | 12.72 | 47.37 | 76.89 | 19.10 |
| learned | 3 | 2 | 256 | 37.00 | 11.62 | 52.82 | **81.10** | 57.84 |
| learned | 1 | 0 | 256 | 21.85 | 20.55 | 34.34 | 78.03 | 53.10 |
| red square | 1 | 0 | 256 | 15.00 | 23.53 | 26.30 | 76.63 | 18.22 |
| green square | 1 | 0 | 256 | 19.91 | 19.61 | 32.73 | 78.31 | 36.53 |

| *TAP-Vid CFG* | AJ↑ | AD↓ | $< \delta_{avg}^x$↑ | OA↑ | OF1↑ |
|---|---|---|---|---|---|
| S RAFT [40] | 69.69 | 1.43 | 83.83 | 81.98 | 46.08 |
| SEA-RAFT [50] | 69.89 | 1.44 | 84.82 | 82.00 | 47.52 |
| DPFlow [33] | 78.09 | 0.99 | 87.86 | 90.19 | 68.57 |
| U SMURF [39] | 65.75 | 2.40 | 79.45 | 82.26 | 42.65 |
| Opt-CWM 175M | 69.53 | **1.19** | 83.15 | **88.85** | 44.17 |
| Opt-CWM Distilled | **74.77** | 1.46 | **85.03** | 88.74 | **55.39** |
| *TAP-Vid First — Main Benchmark* | | | | | |
| S RAFT [40] | 41.77 | 25.33 | 54.37 | 66.40 | 56.12 |
| SEA-RAFT [50] | 43.41 | 20.18 | 58.69 | 66.34 | 56.23 |
| DPFlow [33] | 49.20 | 16.86 | 62.51 | 71.51 | 60.15 |
| U SMURF [39] | 30.64 | 27.28 | 44.18 | 59.15 | 46.91 |
| Opt-CWM 175M | **47.53** | **8.73** | **64.83** | **80.87** | **60.74** |
| Opt-CWM Distilled | 45.55 | 15.68 | 59.36 | 69.27 | 58.41 |

Table 3: **Mask hyperparmeter variants. (Left) At training time.** We train $\Psi^{RGB}$ with non-temporally factored masking policies similar to Video-MAE [41, 47]. The notation of 55-55 indicates 55% of patches are masked in the first frame and 55% are masked in the second frame. Tube masking selects patches at the same spatial location over time, whereas random independently samples patches in each frame. MAE-style masking during training is strictly worse than the temporally-factored masking policy we use as the standard in this paper (shown for reference in the bottom row). All experiments here use 256x256 resolution, MM-3 and MS-2. **(Right) At inference.** We evaluate a 512 resolution $\Psi^{RGB}$ across various masking ratios for the second frame using the MM-3 and MS-2 setting. The standard masking ratio for all results in this work is included as 90% (Ref.) in this table.

| Mask | Train | Test | AJ↑ | AD↓ | $< \delta_{avg}^x$↑ | OA↑ | OF1↑ |
|---|---|---|---|---|---|---|---|
| tube | 55-55 | 0-90 | 23.94 | 15.61 | 36.90 | 72.19 | 52.36 |
| tube | 75-75 | 0-90 | 22.55 | 15.86 | 39.63 | 58.20 | 52.27 |
| tube | 90-90 | 0-90 | 15.23 | 18.57 | 32.12 | 51.98 | 49.20 |
| random | 75-75 | 0-90 | 29.09 | 14.64 | 42.57 | 73.51 | 57.06 |
| random | 75-75 | 0-75 | 34.06 | 12.79 | 47.54 | 76.07 | 60.81 |
| random | 0-90 | 0-90 | 37.00 | 11.62 | 52.82 | 81.10 | 57.80 |

| Masking Ratio | AJ↑ | AD↓ | $< \delta_{avg}^x$↑ | OA↑ | OF1↑ |
|---|---|---|---|---|---|
| 50% | 42.78 | 10.52 | 58.78 | 79.18 | 60.68 |
| 60% | **43.28** | 10.12 | 59.56 | 80.33 | **60.80** |
| 70% | 43.25 | 9.72 | **59.95** | 81.24 | 59.68 |
| 80% | 42.68 | **9.44** | 59.76 | **81.64** | 57.53 |
| 85% | 41.99 | 9.57 | 59.58 | 80.92 | 54.06 |
| 90% (Ref.) | 40.51 | 9.72 | 58.57 | 80.34 | 50.06 |
| 95% | 37.68 | 10.57 | 55.87 | 79.63 | 45.00 |
| 98% | 32.85 | 13.15 | 50.48 | 78.19 | 41.42 |

**Distillation into efficient architectures** RAFT, SEA-RAFT, SMURF, and DPFlow use a highly efficient but special-purpose flow architecture, rather than large general-purpose ViTs [12]. To isolate the effect of this specific architecture design, we train the DPFlow architecture with pseudo-labels generated by Opt-CWM. Specifically, we take a frame pair for each clip, pseudo-label the motion for 1% of the pixels, and train a DPFlow architecture on this pseudo-labeled dataset. We find that this distilled model outperforms the equivalent self-supervised baseline SMURF and is competitive with the supervised techniques (Table 2, right). This outcome pinpoints that the core reason for Opt-CWM's improved performance is our contribution of the novel optimized counterfactual extraction scheme, and the flexible ability to train on unrestricted data that this approach enables, rather than the ViT architecture as such. It is also a practically useful result, since it enables highly efficient inference using the lightweight DPFlow network.

**Limitations and Negative Societal Impact** The main limitation of our work is the high compute requirements and long training times of video vision transformers. These models are also costly at inference time, which can limit deployment in resource-constrained environments. We demonstrate a pathway to resolving this by distilling the trained Opt-CWM into small and efficient architectures.

This limitation also carries a potential negative societal impact due to the significant environmental footprint associated with large-scale training on compute clusters.

## 5 Conclusion & Future Work

We have demonstrated the effectiveness of Opt-CWM in learning motion concepts, achieving state-of-the-art performance on real-world benchmarks. Our paper takes an essential first step in demonstrating the strong quantitative potential of optimized counterfactuals for probing pre-trained video predictors. An important extension of the current work within the domain of scene motion understanding will be to train a multi-frame version of Opt-CWM to create the next generation of scalable self-supervised point trackers, as it has been shown that combining information form three or more frames can help improve flow and occlusion estimation substantially [25]. We plan to scale Opt-CWM's training data, leveraging the wide availability of videos; and explore various alternatives to the masked autoencoder architecture, such as autoregressive or diffusion-based generative video models.

Equally importantly, our twin ideas – parameterizing an input-conditioned counterfactual probe generator and bootstrapping the learning of the probe-generator parameters with an end-to-end sparse prediction task – are task-generic rather than flow-specific. A key next step will thus be to explore the use of these Opt-CWM methods to create self-supervised estimators for a wide variety of visual quantities, including object segments, depth maps, and 3D shape [3, 44].

## Acknowledgments and Disclosure of Funding

This work was supported by the following awards: Simons Foundation grant SFI-AN-NC-GB-Culmination-00002986-05, National Science Foundation CAREER grant 1844724, National Science Foundation Grant NCS-FR 2123963, National Science Foundation Grant RI 2211258, Office of Naval Research grant N00014-20-1-2589, ONR MURI N00014-21-1-2801, ONR MURI N00014-24-1-2748, and ONR MURI N00014-22-1-2740. We also thank the Stanford HAI, Stanford Data Sciences, the Marlowe team, and the Google TPU Research Cloud team for computing support.

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

# A Implementation Details

## A.1 Architecture Details

### A.1.1 $\Psi^{\text{RGB}}$

The input video is first divided into non-overlapping spatiotemporal patches of size $8 \times 8$, with a subset of patches masked. Unlike MAE, we train with both revealed input patches and mask tokens provided to the encoder. We train with the ViT-B architecture [20] with each transformer block consisting of a multi-head self-attention block and an MLP block, both using LayerNorm (LN). The CWM decoder has the same architecture as the encoder. Each spatiotemporal patch has a learnable positional embedding, which is added to both the encoder and decoder inputs. CWM does not use relative position or layer scaling [1, 20]. Please refer to [44, 3] for more details on the architecture. The 175M CWM model is based on ViT-B [12] but has twice the number of total layers. The 1B CWM model is similar to the 175M model, but has 48 layers with an embedding dimension of 2048 and 16 heads.

**Default settings** We show the default pre-training settings in Table 4. CWM does not use color jittering, drop path, or gradient clip. Following ViT's official code, Xavier uniform is used to initialize all transformer blocks. The learnable masked token is initialized as a zero tensor. Following MAE, we use the linear lr scaling rule: $lr = base\_lr \times batch\_size\,/\,256$ [20].

Table 4: **Default pre-training setting of CWM**

| config | value |
|---|---|
| optimizer | AdamW [30] |
| base learning rate | 1.5e-4 |
| weight decay | 0.05 |
| optimizer momentum | $\beta_1, \beta_2 = 0.9, 0.95$ [9] |
| accumulative batch size | 4096 |
| learning rate schedule | cosine decay [29] |
| warmup epochs [16] | 40 |
| total epochs | 800 |
| flip augmentation | no |
| augmentation | MultiScaleCrop [46] |

### A.1.2 $\Psi^{\text{flow}}$

The architecture of the flow-conditioned predictor, $\Psi^{\text{flow}}$, is a vision transformer with 16 layers and 132M parameters. Input images are resized to 224x224, and the patch size is 8. Sinusoidal positional encodings are used. For the encoder, the embedding dimension is 768, and 12 attention heads are used. For the decoder, the embedding dimension is 384, and 6 attention heads are used.

This model has two parallel "streams", the first of which takes RGB input and the second of which takes sparse flow, concatenated with RGB (which is masked to have the same sparsity as the flow), as input. All RGB inputs are from the first frame only; this requires the model to depend solely on flow to modify the RGB and predict the next frame.

The transformer architecture then applies self-attention to each stream and cross-attention between streams. The encoder has 12 layers, split into three groups of 4. In each group, there is one layer with self-attention on each stream and cross-attention from each stream to the other, followed by three layers with only self-attention on the first stream. The decoder has 4 layers; the first applies self-attention to each stream and cross-attention from each stream to the other; the second applies self-attention to the first stream and cross-attention from the second stream to the first; and the final two only apply self-attention to the first stream.

## A.2 Training Details

### A.2.1 $\Psi^{\text{RGB}}$

We train CWM at 256 resolution for 800 epochs and finetune at 512 resolution for 100 epochs by interpolating the positional embeddings. It takes approximately 4 days to train 800 epochs on a TPU

v5-128 pod. We pre-train CWM on the Kinetics-400 dataset [26], without requiring any specialized temporal downsampling.

We train CWM 1B at 256 resolution for 200,000 iterations with a batch size of 512 on a custom video dataset called BVD (approximately equivalent to 400 kinetics epochs). We use the AdamW optimizer, with norm clipping 1.0 and weight decay 0.1. We warmup the learning rate over 2,000 steps to a peak of 3e-4, then linearly decay to 0 over the next 198,000 steps. Training takes approximately 1 day on 64 H100 GPUs.

### A.2.2 $\Psi^{\texttt{flow}}$ and $\texttt{FLOW}_\theta$

We train $\Psi^{\texttt{flow}}$ and $\texttt{FLOW}_\theta$ jointly using an AdamW optimizer with weight decay of $0.05$, betas of $(0.9, 0.95)$, and a learning rate schedule with max learning rate $1.875 \times 10^{-5}$, 40 warmup epochs (10% of total training epochs), and cosine decay. We used a batch size of 32, training on the Kinetics-400 dataset [26].

### A.3 Training Data

We construct a training dataset called BVD (Bid Video Dataset) which consists of publicly available datasets such as Kinetics400 [26] and SomethingSomethingV2 [17] along with other publicly available videos. We filter the videos based on CLIP [36] categories to remove thumbnails and videos with a lot of text in the frame. We additionally filter the videos based on optical flow to remove videos with little motion such as slide shows, or mostly static videos.

### A.4 Inference Techniques

### A.4.1 Multi-Mask

In the process of computing flows in $\texttt{FLOW}_\theta$, at inference time, we take an argmax over the difference between the predicted next frame with and without the counterfactual perturbation. This difference image, $\Delta$, depends on the choice of the random mask as this mask is used by $\Psi^{\texttt{RGB}}$ for the next-frame reconstruction. As discussed in the main text, if a random mask reveals patches too close to where the perturbation should be reconstructed, the predictor $\Psi^{\texttt{RGB}}$ may not reconstruct the perturbation properly, and the difference image will be noisy and diffuse, preventing the model from accurately predicting the next-frame location. Additionally, the reconstructed pixels will not necessarily be the same across different random samplings of visible patches, which may add random noise to the difference image. Both of these issues are ameliorated by our multi-mask technique, in which we compute difference images for a variety of sampled random masks (we found 10 to be a good number of masks for multi-masking), average over the difference images, and then take the argmax of this averaged $\Delta_{\text{avg}}$ to compute next-frame location for determining flow.

### A.4.2 Multiscale

Multiscale refinement of the original flow prediction improves Opt-CWM's performance, as observed in Figure 5. Given an input frame pair and an initial flow prediction, we perform iterative multiscaling through the following procedure. At each "zoom iteration", we take a $0.75H \times 0.75W$ crop of the input frames with original height $H$ and width $W$. We center the second frame crop on the location predicted by the previous iteration.

The transformer-based architecture of the next frame predictor $\Psi^{\texttt{RGB}}$ imposes a limit to the input resolution, which may occasionally prevent small objects or minute features of the input frame from being accurately reconstructed in great detail. Multiscale refinement of the initial flow prediction can be greatly beneficial under these circumstances. However, Figure 5 suggests that the improvement is not monotonic; indeed, excessive cropping may lead to the loss of global context that is necessary to accurately reconstruct the scene. Opt-CWM is run on 4 zoom iterations, which we have empirically found to be optimal.

### A.4.3 Occlusion Estimation

The difference image $\Delta$ can also be used to predict whether a visible point becomes occluded in the next frame. Conceptually, as described in Section 3 in the main text, when a point becomes occluded,

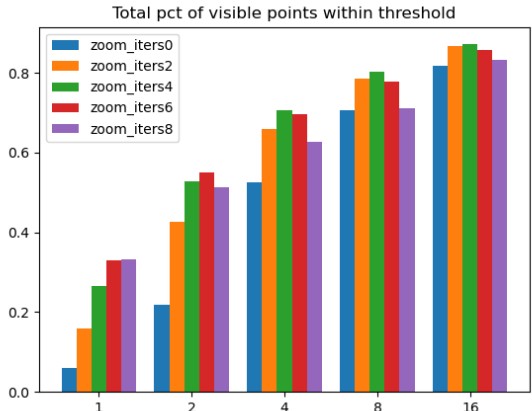

Figure 5: $< \delta_{\mathbf{avg}}$ **broken down across thresholds ($x$-axis).** Fraction of points with error less than a fixed threshold, as a function of number of multiscale (MS) iterations, for pixel thresholds 1, 2, 4, 8, and 16. We find that 4 zoom iterations tends to perform the best, especially for robustness on difficult examples (evidenced by better performance on higher thresholds).

the counterfactual perturbation placed on the object should not be reconstructed in the second frame. Thus, while we take argmax $\Delta$ to compute flow, we can instead use $\max \Delta$ as a signal for occlusion. In particular, we compare $\max \Delta$ to some threshold $t_{\text{occ}}$ to predict occlusion (i.e., we consider the model to have predicted that a point becomes occluded if and only if $\max \Delta < t_{\text{occ}}$).

In the multi-masking setting with 10 masking iterations, we have 10 difference images: $\Delta_1$, $\Delta_2$, ..., $\Delta_{10}$. Instead of thresholding the average, $\Delta_{\text{avg}}$, we can get an improved signal by considering $\max \Delta_i$ for each $i = 1, ..., 10$. In this setting, we found that checking $\frac{1}{10} \sum_{i=1}^{10} \max \Delta_i < 0.05$ provided a good signal for predicting occlusion, and this prediction criterion is what was evaluated in the OA and OF1 metrics of Table 1 in the main paper.

## B  Additional Quantitative Results

### B.1  Constant Frame Gap Protocol

**TAP-Vid Constant Frame Gap (CFG)**. For fair comparison with optical flow models, we also propose an additional protocol with fixed frame gaps that is more advantageous for these baselines (see supplementary for the effect of frame gap on flow baselines). In particular, a fixed 5-frame gap is used: metrics are computed on all frame-pairs that are 5 frames apart (and the point is visible in the first).

We show results for this protocol in Table 5. We observe that in this setting, which is favorable to optical-flow based models, Opt-CWM largely outperforms all unsupervised methods and is competitive with state-of-the-art fully supervised methods.

### B.2  Precision Analysis

Figure 6 attempts to explain the high performance of Opt-CWM on TAP-Vid First through a similar analysis done in Section A.4.2. Our best-performing model (with optimal inference-time configurations) is able to predict the next frame location within 16 pixels of the ground truth for over 85% of the total number of visible points. Unlike baseline models, Opt-CWM is able to predict most points within a reasonable boundary. Further, Opt-CWM predictions are precise; it predicts the majority of the query points within 2 pixels of the ground truth. While SEA-RAFT, which is supervised, is also precise for lower thresholds, the magnitude of the error for wrong predictions is evidently higher, as its performance quickly plateaus for higher thresholds.

As discussed in Section 4 in the main paper, we further evaluate on a custom constant-frame gap protocol (CFG) for fairer comparison with optical flow baselines. As shown here in Figure 7, all models improve significantly under this less challenging setup. In particular, optical flow baselines

Table 5: **Quantitative results on TAP-Vid dataset (Constant five-Frame Gap (CFG).** In the CFG protocol, point tracking is evaluated at fixed gaps of 5 frames, making it an easier setting that is more favorable to optical flow methods. "S" and "U" indicate supervised and unsupervised, respectively. Doduo is not strictly unsupervised as it uses segmentation labels. GMRW is trained on the Kubric dataset, (marked with ‡), making it a more favorable evaluation setting for that method because of the minimal domain gap. Best performing supervised models (shaded) are considered separately.

| | Method | DAVIS | | | | | Kinetics | | | | | Kubric | | | | |
|---|---|---|---|---|---|---|---|---|---|---|---|---|---|---|---|---|
| | | AJ↑ | AD↓ | $<\delta^x_{avg}$↑ | OA↑ | OF1↑ | AJ↑ | AD↓ | $<\delta^x_{avg}$↑ | OA↑ | OF1↑ | AJ↑ | AD↓ | $<\delta^x_{avg}$↑ | OA↑ | OF1↑ |
| S | CoTracker-v3 [25] | 74.49 | 1.21 | 86.59 | 90.74 | 73.08 | 79.45 | 0.82 | 87.86 | 95.65 | 74.12 | 78.15 | 1.02 | 89.07 | 92.59 | 79.92 |
| | RAFT [40] | 69.69 | 1.43 | 83.83 | 81.98 | 46.08 | 79.01 | 0.86 | 87.59 | 92.73 | 49.49 | 73.38 | 1.24 | 83.73 | 91.00 | 63.17 |
| | SEA-RAFT [50] | 69.89 | 1.44 | 84.82 | 82.00 | 47.52 | 75.12 | 1.07 | 85.82 | 88.90 | 39.42 | 77.53 | 1.00 | 87.02 | 92.50 | 68.65 |
| | DPFlow [33] | 78.09 | 0.99 | 87.86 | 90.19 | 68.57 | 80.07 | 0.82 | 87.62 | 95.86 | 75.09 | 87.19 | 0.77 | 93.60 | 93.12 | 79.18 |
| U† | Doduo [23] | 25.61 | 1.61 | 72.56 | 37.49 | 22.59 | 35.26 | 1.19 | 77.62 | 43.00 | 11.63 | 56.57 | 1.74 | 68.63 | 87.26 | 55.01 |
| U | GMRW [37] | 61.28 | 3.11 | 72.28 | 73.01 | 40.31 | 75.44 | 1.23 | 83.54 | 88.89 | 40.96 | 75.54‡ | 1.61‡ | 84.30‡ | 83.92‡ | 53.97‡ |
| | SMURF [39] | 65.75 | 2.40 | 79.45 | 82.26 | 42.65 | 78.76 | 0.97 | 87.16 | 93.13 | 47.69 | 69.05 | 1.59 | 82.38 | 90.84 | 53.49 |
| | CWM [3, 44] | 27.56 | 4.65 | 38.55 | 88.90 | 5.41 | 34.00 | 3.93 | 43.37 | 95.17 | 5.95 | 30.72 | 4.05 | 42.33 | 88.44 | 4.27 |
| | Opt-CWM 1B (ours) | 75.26 | 0.96 | 87.84 | 88.09 | 54.82 | 78.15 | 0.95 | 87.68 | 92.10 | 43.80 | 82.89 | 0.80 | 92.43 | 91.42 | 65.43 |

exhibit strong sub pixel precision. However, we see that in general, compared to self-supervised baselines, Opt-CWM make reasonable predictions of a point's next frame location more often, at a rate comparable to the fully supervised SEA-RAFT.

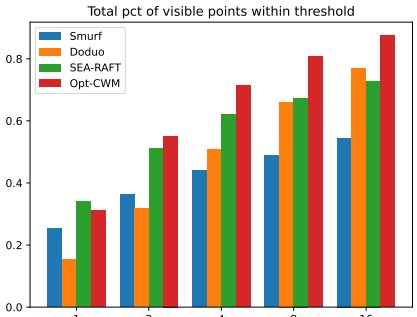

Figure 6: **TAP-Vid First: comparing baseline models on $< \delta_{avg}$ broken down across thresholds ($x$-axis).** Fraction of points with error less than a fixed threshold, as a function of baseline model. Compared to baseline models, Opt-CWM maintains high performance on all thresholds even when making predictions across large frame gaps, as is necessary for TAP-Vid First.

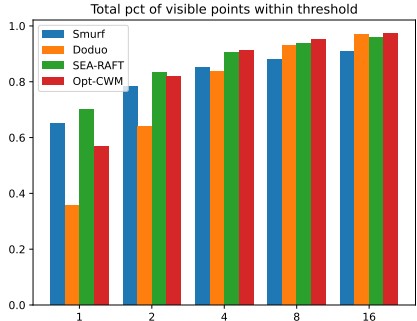

Figure 7: **TAP-Vid CFG: comparing baseline models on $< \delta_{avg}$ broken down across thresholds ($x$-axis).** Fraction of points with error less than a fixed threshold, as a function of baseline model. For fair comparison, we also evaluate on a constant frame gap setting that is more favorable to optical flow baselines. While baseline methods show strong performance for very low thresholds ($<$ 2 pixels), we see that in general Opt-CWM outperforms self-supervised methods and is comparable with SEA-RAFT in predicting more points within a reasonable boundary.

## B.3 Perturbation Across Epochs

The performance of FLOW$_\theta$ is greatly dependent on the quality of its learned Gaussian perturbations. In Figure 8, we see that the appearance of the perturbation evolves alongside the training of Opt-CWM. As the perturbation converges into an optimal patch bespoke for the input frame, the quality of the flow prediction improves in tandem.

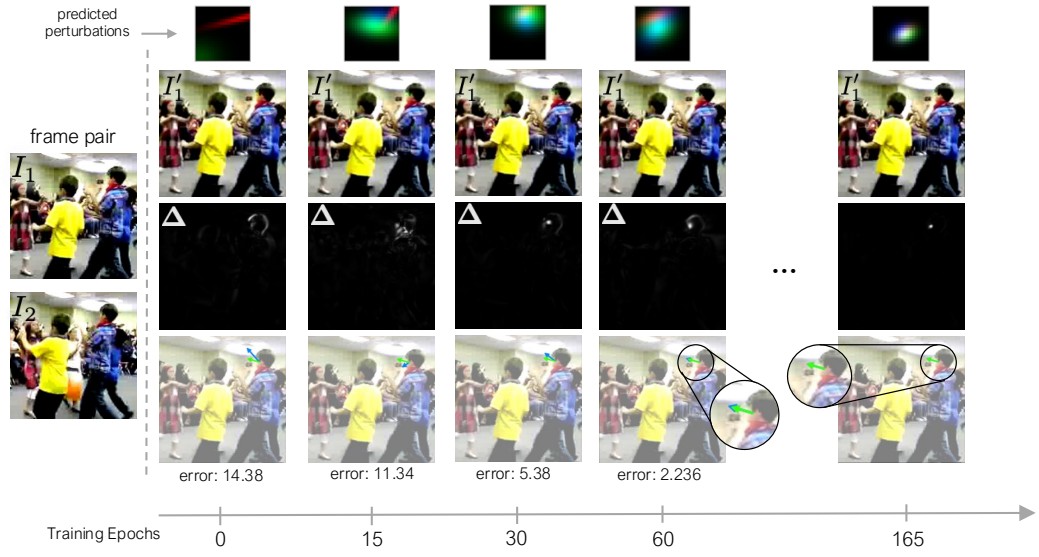

Figure 8: **Evolution of perturbations across training epochs:** We observe how the predicted perturbations change as the model trains. The perturbation starts as a disjoint streak of colors and converges to a localized peak. This in turn increasingly concentrates the difference image $\Delta$ and leads to better flow prediction. Green is the ground truth flow obtained from the TAP-Vid dataset, and blue is our model's prediction.

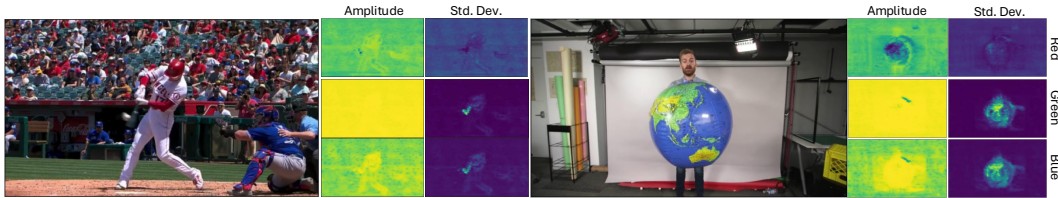

Figure 9: **Perturbation maps reflect scene properties.** For two example frame pairs, we show the amplitudes and standard deviations, at each spatial position and for each color channel, of the optimal Gaussian perturbations predicted by MLP$_\theta$. These "perturbation maps" emergently reflect scene properties, with perturbation parameters varying in size and magnitude depending on where they are located in the image, corresponding to the presence of foreground objects and their parts.

## B.4 Comparison with DINO-Tracker

We compare our results with DINO-Tracker [42], a test-time training approach using pre-trained DINO features that shows promising results on real-world videos. For fair comparison with Opt-CWM and other baselines, we constrain DINO-Tracker, a multi-frame tracker, to run under the same two-frame constraint. Compared with Opt-CWM on TAP-Vid DAVIS, DINO-Tracker obtains an average distance (AD ↓) of 5.91 (vs. 7.70) and a score of 72.17 (vs. 68.63) on the $< \delta^x_{\text{avg}}$ ↑ metric. However, note that DINO-Tracker is not directly comparable as a baseline for Opt-CWM, as it requires per-video test-time optimization and relies on flow predictions from RAFT [40], which is supervised. In contrast to this, Opt-CWM is feed-forward and completely self-supervised.

