# OpenReview forum: "Self-Supervised Learning of Motion Concepts by Optimizing Counterfactuals"
_NeurIPS.cc/2025/Conference — NeurIPS 2025 spotlight_

### Official Review · Reviewer_TsiR · 2025-06-30

**Clarity:** 3
**Significance:** 3
**Originality:** 3
**Rating:** 5
**Confidence:** 4

**Summary:**

In this paper, the authors propose an unsupervised pipeline for training a motion estimator on unlabeled video data. First, learnable probes are inserted into an RGB frame, and a masked autoencoder is employed to predict the next frame based on the unmasked region and the previous frame, enabling the extraction of probe motion. Second, a flow-conditioned next-frame predictor is established to derive a supervision signal from pixel-level loss. Specifically, this predictor generates the subsequent RGB frame conditioned on the previous frame and the estimated flow. This design facilitates gradient backpropagation to both the probe generator and the flow predictor. Evaluated on multiple video datasets, the proposed method achieves consistent and significant improvements.

**Questions:**

1. The next-frame predictor handles temporal dynamics in a 2-frame context; however, there are high-frame-rate and low-frame-rate videos on the internet, and the motion across adjacent frames may have a lot of differences. Can the results of the proposed method be affected by this variance?
2. The training pipeline fully relies on the next-frame prediction world model to generate the warped position of the probes in the masked region of the second frame. What if the unmasked region conflicts with the pasted probes? Does the mask strategy take into account this situation?

**Ethical Concerns:**

["NO or VERY MINOR ethics concerns only"]

**Final Justification:**

According to the summarized pros and cons above, and given that all my concerns about the proposed method are addressed, I recommend an acceptance decision.

**Limitations:**

The training process utilizes a pre-trained world model to generate new positions of the inserted probs and train a flow model based on the updated positions of the probs. This includes two main limitations:
- Firstly, the quality of the generated next frame depends on the pre-trained world model. Once its capacity can not match the scale of the training data, the flow model performance may be hindered by the sub-optimal synthesized frames.
- Secondly, to calculate the position of the probs in the next frame, the authors apply a heuristic method that calculates the difference image and finds its peak. This means the inserted probs should be sparse and may be affected by occlusion of different probs; thus, the number of reference points can not be scaled up in an example. Although the proposed training pipeline neglects the number of available probs, it may still face the challenge of training efficiency.

**Quality:**

3

**Strengths And Weaknesses:**

**Strengths:**
- The proposed Opt-CWM methods yield significant improvement over previous methods and even beat supervised methods across three different video datasets, which reveals their effectiveness and advancedness.
- The distilled SEA-RAFT model from Opt-CWM using unsupervised data shows competitive results against the supervised learned SEA-RAFT variant, which shows a promising ability to learn robust motion representation from internet-scale videos.
- All proposed modules (probe generator, flow function, flow-based next frame predictor) in the paper are trained unsupervised, without separate, direct supervision, yet they mutually reinforce each other and collectively enable robust motion extraction.

**Weakness:**
- Learning motion clues from unlabed video is similar to the previous idea [1], I recommend that the authors cite and discuss this literature in the related work section.
- The proposed parameterized counterfactual probe generator is trained in an unsupervised manner without direct supervision, which means the generalization ability of the probe generator is still restricted by the heuristic used to generate the training data. Since the probe generation process is similar to the image editing task, the authors can try to use off-the-shelf image editing methods like [2] to generate probes.

[1]. Masked Motion Encoding for Self-Supervised Video Representation Learning.
[2]. LEDITS++: Limitless Image Editing using Text-to-Image Models.

---

> ### Author Rebuttal · Authors · 2025-07-31
>
> **R1 Q1**: *Learning motion clues from unlabed video is similar to the previous idea [1], I recommend that the authors cite and discuss this literature in the related work section.*
>
> **R1 A1**:  Thank you for the suggestion. We will include a discussion of [1] in the related work section. That work focuses on learning video representations for video action recognition by reconstructing motion trajectories and object shapes from sparse video patches, using ground-truth motion trajectories extracted via traditional optical flow methods.
>
> By contrast, our work serves a different purpose: we are not aiming to learn a video representation for video action recognition. Instead, we propose a method to extract optical flow signals from a world model that has already been trained to predict future RGB frames. Our approach is self-supervised in the sense that it does not require external flow annotations or supervision. In fact, the flows extracted via our method could be used as training data for models like [1], providing a viable alternative to using other available off-the-shelf flow algorithms.
>
>
> ---
>
> **R1 Q2**: *The proposed parameterized counterfactual probe generator is trained in an unsupervised manner without direct supervision, which means the generalization ability of the probe generator is still restricted by the heuristic used to generate the training data. Since the probe generation process is similar to the image editing task, the authors can try to use off-the-shelf image editing methods like [2] to generate probes.*
>
> **R1 A2**: Thank you for the suggestion regarding the use of off-the-shelf image editing methods such as [2]. However, we would like to clarify that our probe generation process is fundamentally different from image editing, particularly from text-driven editing pipelines like [2]. Our method does not aim to generate semantically meaningful image edits. Instead, we apply a small, image-conditioned, localized perturbation (a learnable patch) to the input frame and analyze how this perturbation propagates across time through the model's predicted next frame. This allows us to estimate the local optical flow.
>
> ---
>
> **R1 Q3**: *Firstly, the quality of the generated next frame depends on the pre-trained world model. Once its capacity can not match the scale of the training data, the flow model performance may be hindered by the sub-optimal synthesized frames.*
>
> **R1 A3**: Thank you for your comment. We agree that the quality of the generated next frame depends on the capacity of the underlying world model. As noted in prior work on CWM [2, 23] as well as results in our paper, increasing model capacity consistently improves performance on flow estimation. While it is possible that such gains may eventually saturate, current results suggest we have not yet reached that point.
>
> We therefore believe it is premature to conclude that motion estimation will be fundamentally limited by the fact that the model's performance may saturate at larger scales. Rather, we view this as an open question for future work. For now, our approach is justified by the empirical improvements observed at the scales at which the current CWM model is trained.
>
> ---
>
> **R1 Q4**: *Secondly, to calculate the position of the probs in the next frame, the authors apply a heuristic method that calculates the difference image and finds its peak. This means the inserted probs should be sparse and may be affected by occlusion of different probs; thus, the number of reference points can not be scaled up in an example. Although the proposed training pipeline neglects the number of available probs, it may still face the challenge of training efficiency.*
>
> **R1 A4**: We agree with the reviewer that inserting multiple probes into the same image can lead to interference effects such as occlusion and overlapping influence regions, which makes simultaneous probing challenging. However, our current implementation addresses this by applying a single perturbation per image and executing all probes in a batched and parallelized manner. This allows for efficient extraction of flow signals without compromising the integrity of individual probes.
>
> Furthermore, once we extract sparse flow supervision from the model using these probes, we can train a separate supervised optical flow estimator (e.g., SEA-RAFT) on this data. As shown in our paper (Table 2, right), these distilled models are not only significantly faster at inference time, but also achieve good performance, demonstrating that our approach is practical despite the sparsity of individual probe. We summarize these results below, and present an additional experiment where we distill Opt-CWM into the newer DPFlow [A] model, which achieves better performance.
>
> | Model | AJ | AD | Pct | OA | OF1 |
> |  :--- | :---: | :---: | :---: | :---: | :---: |
> | RAFT | 41.77 | 25.33 | 54.37 | 66.40 | 56.12 |
> | SEA-RAFT | 43.41 | 20.18 | 58.69 | 66.34 | 56.23 |
> | SMURF | 30.64 | 27.28 | 44.18 | 59.15 | 46.91|
> | Opt-CWM 175M | 47.53 | 8.73 | 64.83 | 80.87 | 60.74 |
> | SEA-RAFT (distilled) | 44.05 | 17.49 | 57.93 | 69.75 | 60.72 |
> | DPFlow (distilled) | 45.55 | 15.68 | 59.36 | 69.27 | 58.41 |
>
> [A] DPFlow: Adaptive Optical Flow Estimation with a Dual-Pyramid Framework - Henrique Morimitsu, Xiaobin Zhu, Roberto M. Cesar Jr., Xiangyang Ji, Xu-Cheng Yin

---

> > ### Comment · Reviewer_TsiR · 2025-08-05
> >
> > Thanks to the authors for their replies. I have no further concerns.

---

### Official Review · Reviewer_5xm4 · 2025-07-03

**Clarity:** 2
**Significance:** 3
**Originality:** 3
**Rating:** 4
**Confidence:** 2

**Summary:**

This paper proposes Opt-CWM, a novel self-supervised approach for motion concept learning from videos. By leveraging large-scale pretrained video prediction models, the authors introduce optimized, learnable counterfactual probes that extract high-quality optical flow and occlusion information from real-world videos without any labeled data. The core innovations are: (1) parameterizing the counterfactual probe generator with a neural network, allowing probes to adapt to image content and context, and (2) a self-supervised learning objective that trains the probe generator via a sparse prediction information bottleneck, eliminating the need for heuristic losses or supervision.

**Questions:**

1. Could the authors provide more detailed analysis or ablation experiments showing how the model distinguishes between perturbations that correspond to moving/visible objects versus those that are static or occluded? Is there a theoretical or empirical criterion to ensure that significant model output changes are reliably associated with true object motion, and minimal changes indicate occlusion or background?
2. The current layout places important figures (e.g., Figures 1 and 2) far from their detailed descriptions in the main text, which hinders readability. I recommend the authors to consider reorganizing the paper so that figures and their explanations appear together.
3. Figure 2 does not fully reflect the described functional form in the main paper. In addition, it omits some input variables (such as $I_2$, which may cause confusion about the model’s actual input-output structure.
4. Does Opt-CWM achieve true optimality in generating counterfactual probes, or are there cases where it may still be suboptimal? Is there any analysis or discussion of the theoretical limits of the proposed approach?
5. What is the computational complexity of Opt-CWM compared to existing baselines? Is the method practical for large-scale or real-time applications?
6. Can the authors provide a detailed explanation of all symbols and notations used in the tables, to enhance readability and reproducibility?

**Ethical Concerns:**

["NO or VERY MINOR ethics concerns only"]

**Final Justification:**

Thank you for addressing my questions. I have increased my overall rating for the paper.

**Limitations:**

Yes

**Quality:**

3

**Strengths And Weaknesses:**

Strengths: This paper introduces a novel approach to self-supervised motion and occlusion estimation by leveraging optimized counterfactual probes, which are parameterized by a neural network and learned entirely without supervision. Unlike traditional CWM that relies on fixed, hand-crafted perturbations, Opt-CWM framework utilizes learnable, adaptive perturbations that are dynamically generated based on image content and context. The method achieves state-of-the-art empirical performance on several challenging benchmarks, consistently outperforming both established supervised methods and several self-supervised baselines.

Weaknesses
1. The paper lacks a clear and principled explanation for how the model determines whether an added perturbation will be meaningfully carried forward with the moving object in the model’s prediction.
2. The manuscript suffers from poor layout and figure placement. Important figures (e.g., Figure 1 and Figure 2) are physically distant from their detailed explanations, making it difficult for readers to follow the narrative and connect visuals with the text.
3. While the paper points out that previous CWM methods are suboptimal due to their fixed, hand-crafted perturbations, it does not provide sufficient discussion or evidence on whether Opt-CWM achieves optimality or merely improves upon prior suboptimal solutions.
4. The paper does not provide an analysis of the computational complexity for either training or inference.
5. Symbols and formatting (e.g., underlines, arrows) in the result tables are not explained, which could cause confusion for readers unfamiliar with these conventions.

---

> ### Author Rebuttal · Authors · 2025-07-31
>
> **R3 Q1**: *Could the authors provide more detailed analysis or ablation experiments showing how the model distinguishes between perturbations that correspond to moving/visible objects versus those that are static or occluded? Is there a theoretical or empirical criterion to ensure that significant model output changes are reliably associated with true object motion, and minimal changes indicate occlusion or background?*
>
> **R3 A1**:  To clarify, we briefly re-state the key “theory” behind our approach. Our counterfactual compares the predictor’s next‑frame output with vs. without a small, image‑conditioned perturbation localized at the query pixel in the first image. The resulting difference image $\Delta$ reveals where and how strongly the model carries information from that pixel into the next frame.
>
> This concept leads to different behavior for different points, depending on the condition:
> - **If the point is visible and moving...** then the perturbation appears at a new location in the next frame; $\Delta$ exhibits a **sharp, high‑magnitude peak** with **some displacement** from original coordinates.
> - **If the point is visible and static...** then the perturbation appears in the same position in the next frame; $\Delta$ has **sharp, high magnitude peak** but with **near‑zero displacement**.
> - **If the point becomes occluded/out‑of‑view...**  then the model cannot “carry” the perturbation forward; $\Delta$ has **low magnitude** and a **diffuse response** indicating occlusion.
>
> This behavior is agnostic to foreground/background: background points can move under camera motion (and we detect that as a displacement). We would like to emphasize that we *do not* claim that minimal change is equivalent to being on the background. Minimal change indicates occlusion/out‑of‑view rather than background.  But our motion method works both when the motion is due to camera motion (background) or object motion (foreground): the same calculation is accurate either way.
>
> **R3 Q1(b)**: *The reviewer asks about an **empirical criterion***
>
> **R3 A1(b)**: Regarding an empirical criterion, the evaluation of Occlusion Accuracy (OA) and Occlusion F1 (OF1) already establishes this. We determine whether a point is occluded based on the magnitude of the $\Delta$ response, and observe competitive occlusion estimation performance on TAPVid First (DAVIS: OA 80.44, OF1 68.43; Kinetics: OA 80.74, OF1 76.21; Kubric: OA 90.11, OF1 72.56).
>
> **R3 Q1(c)**: *The reviewer also asks about ablation experiments.*
>
> **R3 A1(c)**: Our ablations also show that learned counterfactual probes help: we obtain better occlusion separation than fixed patches: OA 80.34 / OF1 50.06 (learned) vs. 75.38 / 27.21 (red square) and 76.89 / 19.10 (green square) on the 512px with MM‑3/MS‑2 setting.
>
> ---
>
> **R3 Q2 & Q3**: *The current layout places important figures (e.g., Figures 1 and 2) far from their detailed descriptions in the main text, which hinders readability. I recommend the authors to consider reorganizing the paper so that figures and their explanations appear together. Figure 2 does not fully reflect the described functional form in the main paper. In addition, it omits some input variables (such as, which may cause confusion about the model’s actual input-output structure.*
>
> **R3 A2 & A3**: Thank you for this suggestion. We will revise the organization of the paper and update Figure 2 according to this recommendation to help with readability.
>
> ---
>
> **R3 Q4**: *Does Opt-CWM achieve true optimality in generating counterfactual probes, or are there cases where it may still be suboptimal? Is there any analysis or discussion of the theoretical limits of the proposed approach?*
>
> **R3 A4**: This is an important point. We want to emphasize that we do not make any tight mathematical claims in our work. Our method has an intuitively clear idea for how to optimize the parameters of the function that predicts image-conditioned probes, based on the downstream flow-conditioned reconstruction objective of $\Psi^{\texttt{flow}}$.  This is a principled surrogate for learning counterfactual probes that result in accurate motion estimates. Intuitively it makes sense why it should work, even though we don’t have a math proof.
>
> With our joint optimization approach, probe estimation and sparse flow-conditioned next-frame reconstruction are coupled. While we do not have a formal lower bound of the coupling strength, Figure 3B directly examines the coupling, and finds that it is sufficiently strong to enable optimization of the probes.
>
> And ultimately this concept pays off:  while we cannot formally demonstrate global optimality, our learned image-conditioned probes consistently outperform hand-designed alternatives (Table 3 left).
>
> ---
>
> **R3 Q5**: *Can the authors provide a detailed explanation of all symbols and notations used in the tables, to enhance readability and reproducibility?*
>
> **R3 A5**:
> We provide information on the symbols used in our tables:
> - S, U$^\dagger$, U: Supervised, semi-supervised, self-supervised, respectively. U$^\dagger$ is reserved for Doduo, which uses segmentation labels and thus is not strictly self-supervised.
> - $\textbf{bold}$, $\underline{\text{underline}}$: 1st and 2nd in performance, respectively.
> - shaded: Best performing supervised models are considered separately.
> - $\ddagger$: Trained on Kubric. This is to separate GMRW results on Kubric.
> - AJ, AD, $<\delta^x_\textrm{avg}$, OA, OF1: TAP-Vid metrics. Please refer to Section 4 for details on each metric.
> - MM, MS: Multi-mask and multi-scale, which are inference-time enhancements. Please refer to Section 3.2 for details.

---

> > ### Comment · Reviewer_5xm4 · 2025-08-06
> >
> > Thank you for addressing my questions. I have increased my score by 1.

---

### Official Review · Reviewer_YyNn · 2025-07-03

**Clarity:** 3
**Significance:** 3
**Originality:** 3
**Rating:** 5
**Confidence:** 3

**Summary:**

This paper proposed a new self-supervised learning based method, Opt-CWM, for learning motion concepts. It solves the limitations of an existing self-supervised learning based method method, CWM, in this field with two innovations. First, it propose to parameterize a counterfactual probe policy generator with a learnable neural network that can predict situation-specific probes that take into account the appearance context (both local and global) around target points to be tracked, and thus can be less out-of-distribution than hand-coded probes. Second,  it proposes to learn the probe generator in a principled fashion without relying on any supervision from labeled data or heuristics. Experiments shows the proposed method significantly improves performance of multiple motion concept prediction tasks.

**Questions:**

See the weakness above.

**Ethical Concerns:**

["NO or VERY MINOR ethics concerns only"]

**Limitations:**

Yes.

**Quality:**

3

**Strengths And Weaknesses:**

Strengths
- The proposed method is technically sound.
- Experiments show significant improvement over existing methods.
- The proposed approach is well motivated.

Weakness
- The proposed method uses two large base predictors, one with 175M and the other 1B learnable parameters, which makes the comparison not very fair over the existing methods.
- Due to the introduce of the large models, the inference time of the proposed method should be much higher than the existing methods. It would be great to see the comparison on this axis.
-  To address the limitation on high inference time, it is stated that "We demonstrate a pathway to resolving this by distilling the trained Opt-CWM into small and efficient architectures". However, I do not see such results.
- What is the impact of the scale of the base predictors? Currently two models of 175M and the other 1B are employed. How about using smaller or even larger models? It would be great the see the trend of the performance with respect to the model size.
- Many sentences are too long and complex, making the paper hard to read and follow.

---

> ### Author Rebuttal · Authors · 2025-07-31
>
> **R2 Q1**: *The proposed method uses two large base predictors, one with 175M and the other 1B learnable parameters, which makes the comparison not very fair over the existing methods.*
>
> **R2 A1**:  We acknowledge that the capacity of our model differs from some baselines. Our base CWM is a general‑purpose video prediction foundation model (used here to extract motion, but can also be used to extract depth/segments), which justifies the larger parameter count. Regarding fairness at inference time, we follow a standard approach: distill Opt‑CWM into a small, task‑specific optical‑flow model. Concretely, Table 2 (right) shows that SEA‑RAFT distilled from Opt‑CWM (trained only on pseudo‑labels, no ground‑truth flow) outperforms SMURF (self‑supervised) and is competitive with supervised RAFT, while running at the small‑model speed. We summarize these results below, and present an additional experiment where we distill Opt-CWM into the newer DPFlow [A] model, which achieves better performance. It is also worth noting that this distillation procedure can also be further tuned and result in higher final performance. We will communicate this more directly in the paper and discuss compute cost details.
>
> | Model | AJ | AD | $<\delta_\text{avg}^x$ | OA | OF1 |
> |  :--- | :---: | :---: | :---: | :---: | :---: |
> | RAFT | 41.77 | 25.33 | 54.37 | 66.40 | 56.12 |
> | SEA-RAFT | 43.41 | 20.18 | 58.69 | 66.34 | 56.23 |
> | SMURF | 30.64 | 27.28 | 44.18 | 59.15 | 46.91|
> | Opt-CWM 175M | 47.53 | 8.73 | 64.83 | 80.87 | 60.74 |
> | SEA-RAFT (distilled) | 44.05 | 17.49 | 57.93 | 69.75 | 60.72 |
> | DPFlow (distilled) | 45.55 | 15.68 | 59.36 | 69.27 | 58.41 |
>
> [A] DPFlow: Adaptive Optical Flow Estimation with a Dual-Pyramid Framework - Henrique Morimitsu, Xiaobin Zhu, Roberto M. Cesar Jr., Xiangyang Ji, Xu-Cheng Yin
>
> ------
>
> **R2 Q2**: *Due to the introduce of the large models, the inference time of the proposed method should be much higher than the existing methods. It would be great to see the comparison on this axis.*
>
> **R2 A2**: The size of the base predictor model does in fact lead to slower inference speed. We describe our distillation-based solution to this added inference time cost in **R2 A1** which will have the same inference-time speed as standard optical flow methods. We commit to adding inference time details in the final version.
>
> ------
>
> **R2 Q3**: *To address the limitation on high inference time, it is stated that "We demonstrate a pathway to resolving this by distilling the trained Opt-CWM into small and efficient architectures". However, I do not see such results.*
>
> **R2 A3**: Thank you for raising this, those results are in Table 2 (right) of the main text, but we realize our text could emphasize them better. Please also refer to our discussion in **R2 A1**.
>
> ------
>
> **R2 Q4**: *What is the impact of the scale of the base predictors? Currently two models of 175M and the other 1B are employed. How about using smaller or even larger models? It would be great to see the trend of the performance with respect to the model size.*
>
> **R2 A4**: We acknowledge that establishing scaling laws for base predictors is an important topic. Within our compute budget, we choose to train and evaluate at two orders of magnitude (175M, 1B) and show improvements of  47.53 $\to$ 51.88  for AJ (higher is better) and 8.73 $\to$ 7.70 for AD (lower is better).  A complete scaling study (smaller/larger CWMs) is a promising direction and we plan to include it in future work.
>
> ------
>
> **R2 Q5**: *Many sentences are too long and complex, making the paper hard to read and follow.*
>
> **R2 A5**: Thank you for this helpful feedback, we will revise the draft for clarity and improve the sentence length and structure.

---

### Official Review · Reviewer_ix1k · 2025-07-23

**Clarity:** 3
**Significance:** 2
**Originality:** 4
**Rating:** 5
**Confidence:** 3

**Summary:**

This work explores a new idea for how to extract motion-primitives from large pretrained video models, specifically focusing on optical-flow and occlusion. The authors develop a method Opt-CWM which estimates optical flow and occlusion from pretrined video prediction models by leveraging counterfactual probes, which are input-conditioned perturbations to the appearance or position of visible patches. The authors develop a method for generating these probes in an optimal manner and show that the probe-generation parameters can be learned with a self-supervised method. Given, this Opt-CWM is shown to be SoTA in motion estimation.

**Questions:**

See above weaknesses if these are addressed I'm willing to raise my score.

**Ethical Concerns:**

["NO or VERY MINOR ethics concerns only"]

**Final Justification:**

The authors sufficiently addressed my concerns in the paper and made key clarifications that now highlight the impact of their results as being relatively significant in context of prior work. Given similar addressing of the concerns of other reviewers I believe this work now meets the criteria for a solid acceptance (5) as it is novel and has the potential to lead to advancement in the field of SSL-based video representation learning.

**Limitations:**

yes

**Paper Formatting Concerns:**

No issues.

**Quality:**

3

**Strengths And Weaknesses:**

Strengths
1. The paper is well-written and clear on the contributions.
2. The method for learning optimiized counterfactuals via using the flow prediction as a bottleneck is interesting and novel to my knowledge and approaches this problem in a different way than many existing methods.
3. The ablations are quite ocmprehensive and appreciated.

Weaknesses
1. The authors state that most recent works in point-tracking are semi-supervised (like BootsTAP, TAPNext etc.) and thus are out of scope for comparison. While I appreciate this, I feel it would be disingenuous not include some of these results in the paper for comparison to show where the SoTA is for these tasks. Methods like BootsTAP or TAPNext are extremely scalable because they use pseudo-labeling. Therefore, one might ask if the goal is to solve point-tracking through occlusions etc. generally then what is the inherent benefit of using something self-supervised if the performance is still drastically worse than those methods (see numbers in TAPNext paper Table 1 for example)? The main driver of self-supervised learning to me is it's generality and scalability to places where finding supervision is hard or not clean, but if semi-supervised learning can bridge this gap effectively should these tasks (e.g. TAP-Vid) be the main demonstration of the benefits of this approach?
2. The authors miss some key citations of work that needs to be compared to. For example DINO-Tracker from Tumanyan, Singer et al (ECCV 2024) is also self-supervised and seems to present quantitative results that are far better than what is reported in this paper. Can the authors include this comparison or comment on why it is not necessary?
3. There are some discrepancies in the numbers reported in this work vs. the literature. For example for DAVIS AJ and delta_avg, the authors report GMRW as 36.47 and 54.59. However in the GMRW paper they report: 41.8 and 60.9 . These kinds of discrepancies should be clarified in the work otherwise comparison across works becomes quite tricky. Is there something about the evaluation methodology used that differs from prior work for example?

---

> ### Author Rebuttal · Authors · 2025-07-31
>
> **R1 Q1**: *The authors state that most recent works in point-tracking are semi-supervised (like BootsTAP, TAPNext etc.) and thus are out of scope for comparison. While I appreciate this, I feel it would be disingenuous not include some of these results in the paper for comparison to show where the SoTA is for these tasks. Methods like BootsTAP or TAPNext are extremely scalable because they use pseudo-labeling. Therefore, one might ask if the goal is to solve point-tracking through occlusions etc. generally then what is the inherent benefit of using something self-supervised if the performance is still drastically worse than those methods (see numbers in TAPNext paper Table 1 for example)? The main driver of self-supervised learning to me is it's generality and scalability to places where finding supervision is hard or not clean, but if semi-supervised learning can bridge this gap effectively should these tasks (e.g. TAP-Vid) be the main demonstration of the benefits of this approach?*
>
> **R1 A1**: We agree that this is important to achieve good results when compared to methods trained with more supervision. We believe we have achieved this, although it may be confusing since multiple protocols exist in the literature, e.g. Table 1 in the TAPNext paper reference uses a different (and somewhat easier) protocol than the one we report. On the standard but more relatively challenging benchmark that we report  (please refer to evaluation details below), our method is SoTA, even compared to the semi-supervised methods like CoTracker, and performance numbers for all methods are lower overall.
>
> *Why self-supervised?* Despite the recent strong results of semi-supervised learning, it is still fundamentally limited because it leads to a limited exploration of the possible label space of point tracks. Specifically, pseudo-labeling will further reinforce what the model can already predict confidently. Parts of the data distribution that are difficult to synthesize with graphics rendering engines (and therefore will not be in the initial training dataset before any pseudo-labeling is done) will not be learned with pseudo-labeling. Examples include non-rigid deformable motions (e.g., fabric, dough), or volumetric/particle motion (e.g., liquids, smoke, dust). Self-supervised learning through predictive modeling, like the Counterfactual World Modeling paradigm that we build upon in this paper does not have this limitation of pseudo-labeling and can learn such challenging scenarios from data.
>
> *Evaluation details*: We follow the more challenging “first” protocol for TAPVid evaluation, whereas the TAPNext paper follows the easier “strided” protocol, leading to higher reported tracking performance. Briefly, in the “first” protocol, points are tracked from the first time they appear through the end of the video. In the "strided" protocol, a point is queried every 5 frames if it is visible and tracked forward and backward in time. First is more challenging because on average points are tracked over longer temporal frame gaps. For more information refer to Appendix H of the original TAPVid paper. We also run all models on pairs of query and target frames to track a point through a video. This is (1) to remove the effect of additional algorithmic heuristics on how to chain frame-pair predictions into tracks and to reveal the capability of the underlying motion representation, and (2)  because this represents a fundamental unit of the problem of motion understanding–tracking directly emerges from estimating the motion/occlusion of a point across two frames
>
> ---
>
> **R1 Q2**: *The authors miss some key citations of work that needs to be compared to. For example DINO-Tracker from Tumanyan, Singer et al (ECCV 2024) is also self-supervised and seems to present quantitative results that are far better than what is reported in this paper. Can the authors include this comparison or comment on why it is not necessary?*
>
> **R1 A2**:  We thank the reviewer for raising this important prior work, we will make sure to add this paper to the related work. Regarding a comparison, DINO-tracker is not a strictly self-supervised method because it assumes an off-the-shelf RAFT optical flow model to derive pseudo-labels, which has been trained with explicit optical flow supervision. In contrast, our approach is only supervised by a next frame pixel prediction loss. We will include a comparison with DINO tracker in the final version.
>
> ---
>
> **R1 Q3**: *There are some discrepancies in the numbers reported in this work vs. the literature. For example for DAVIS AJ and delta_avg, the authors report GMRW as 36.47 and 54.59. However in the GMRW paper they report: 41.8 and 60.9 . These kinds of discrepancies should be clarified in the work otherwise comparison across works becomes quite tricky. Is there something about the evaluation methodology used that differs from prior work for example?*
>
> **R1 A3**:  We thank the reviewer for raising this point. As in the **R1 A1** response, we would like to clarify that in this paper we follow the more challenging “first” protocol for TAPVid evaluation, whereas the GMRW paper follows the simpler “strided” protocol, leading to higher reported tracking performance. As discussed, our more challenging evaluation protocol evaluates models' capabilities at the fundamental problem of estimating the motion of a point across a pair of frames.

---

> > ### Comment · Reviewer_ix1k · 2025-08-05
> > **Response to rebuttal**
> >
> > I thank the authors for their rebuttal:
> > A1- I appreciate the cited benefits of self-supervised methods in general but I have an additional clarification regarding the "first" vs. "strided" protocol the authors are claiming is the difference. In the TAPNext paper Table 1 they clearly have results for both DAVIS "first" and DAVIS "strided" as well as Kinetics "first" and Kinetics "strided" for DAVIS "first" and Kinetics "first" their results are worse than the "strided" eval as you correctly point out, but they are still far above your results with models that have far fewer learnable params (e.g. ViT-B vs. your 1B). Furthermore, if you look at the Table 3 caption from the TAPNext paper they even explicitly say "We evaluate every ablation on DAVIS query-first." In this table, their DAVIS query-first eval is showing 55.0 AJ with a small-scale ViT-S run (still higher than your 51.88 AJ with your method and 1B model). Similar results that are better than your reported numbers can be found in the BootsTAP paper as well which use the query-first evaluation. As a result, I still think my concern regarding SoTA results stands.
> >
> > A2 thank you for the inclusion - this addresses my concern.
> >
> > A3 thank you for the response- I have reviewed the paper and agree; however, see above with my concerns regarding the semi-supervised methods like TAPNext and BootsTAP.
> >
> > I appreciate the response from the authors; however, unless I am misunderstanding the prior work that is semi-supervised, they do in fact use the query-first evals and are significantly better in performance on the TAPvid benchmark. If the authors can clarify further and prove to me that their numbers are in fact SoTA even compared with these methods then I will consider raising my score but without that, I will maintain my score.

---

> > > ### Author Response · Authors · 2025-08-06
> > >
> > > We appreciate the reviewer’s follow‑up and believe that we now better understand the source of confusion. The distinction is not so much about “first” vs. “strided”, but whether the protocol uses all the frames between two timepoints or just the pair of frames at the two time points as model input. There is a version of the First protocol that just uses two frames, and one that feeds the algorithm all the intermediate frames between the two time points. The two-frame version is meaningfully more difficult than the multi-frame one because it gives less information to the algorithm. This represents the "core dynamics understanding" version of the point tracking task: how a point will move (or become occluded) across two frames. The multi-frame version of this problem is easier because it allows for aggregating multiple comparisons across a sequence of frames (different algorithms implement this part differently), and therefore smoothes out some of the hard cases.
> > >
> > > Doing a two-frame evaluation like we adopt in this work is a "standard thing" to do in the optical flow community, whereas the multi-frame evaluation is the "standard thing" to do in the tracking community. These distinctions are due to the different goals of the works (optical flow and tracking models serve different purposes). In this work, we aim to contribute to the optical flow domain but evaluate on challenging and diverse real world data.
> > >
> > > We have evaluated a strong supervised tracking baseline (CoTracker v3) in this two-frame context, and observe that the numbers drop substantially, which we believe is expected because this two-frame setting is harder. What we conclude from that is that significant gains come from using the additional information in the intermediate frames, rather than improved dynamics understanding in the fundamental two-frame case. This is a fair comparison as it allows us to put that method on an equal footing with all the other (naturally two-frame) supervised and self-supervised optical flow methods that we investigate in this paper (not just Opt-CWM).
> > >
> > > Our baselines are very recent supervised state-of-the-art flow models. While our method is better than those, those methods are also better than CoTrackerV3 at the two-frame evaluation, which is evidence that the "core two-frame temporal correspondence" problem is a meaningful target and where our method performs well.
> > >
> > > In the long run, we believe that improved optical flow methods will subsequently lead to better multi-frame models (e.g. some initial point tracking techniques like PIPS meaningfully borrow from RAFT). We believe that our approach can have meaningful extensions to a multi-frame setting as well, that will lead to strong performance, which are exciting directions for future work.

---

> > > > ### Comment · Reviewer_ix1k · 2025-08-06
> > > > **Response to authors**
> > > >
> > > > I thank the authors for the detailed explanation- after consideration of this I am willing to raise my score conditioned on the authors modifying the text in the paper to highlight this distinction more clearly in the evaluation section. I just re-read the paper and while it is mentioned in section 4, it is very hard to discern that this difference makes such a big difference compared with the protocols used in all of the prior tracking work. It is natural for people to compare these tables to prior work and it should be very clear in the evaluation protocol that this is what has been done along with an explanation for why.
> > > >
> > > > Nevertheless, I now understand the distinction and feel the results are strong enough to warrant acceptance.

---

> > > > > ### Author Response · Authors · 2025-08-07
> > > > > **Official Comment by Authors**
> > > > >
> > > > > We thank the reviewer for the fruitful discussion! We will take the reviewer's advice and make the evaluation protocol, as well as the important points raised in this thread, clearer in the main text.

---

### Note · Authors · 2025-08-15

We thank the reviewers for their constructive feedback and engaging discussion. We are grateful that the reviewers found our work novel, technically sound, well motivated, and yielding significant improvements.

In response to the discussion, we will make sure to clearly explain and justify our two-frame evaluation protocol in the draft so it will not be missed by the reader. We will also better highlight our distillation results, showing that Opt-CWM can be a source of supervision for fast and light-weight architectures while still maintaining high performance. Finally, we will incorporate all feedback on clarity and readability.

---

### Decision · Program_Chairs · 2025-09-17

**Decision:**

Accept (spotlight)

**Comment:**

This paper considers the task of extracting optical flow and occlusion from large pretrained video models using counterfactual probes: input-conditioned perturbations to the appearance of image regions. On the whole, the reviewers consider this a technically sound and well-motivated approach that elicits significant improvements in practice. The identified strengths include the design choices, such as the idea of using a probe generator that takes into account local and global appearance context so that the probes are more in-distribution than a hand-coded probe, the principled self-supervised learning objective, and the use of flow prediction as a bottleneck, as well as the performance that is competitive with supervised approaches. The identified weaknesses included concerns about evaluation settings (satisfactorily clarified in the author response), model capacity (likewise, via a distillation approach), the optimality claim (clarified to be meant in a loose, rather than mathematical sense), and presentation/readability/clarity.

The initial ratings skewed positive with some deviation between reviewers. Upon reading all reviews, author responses, and engaging in discussion with the authors, the reviewers formed a consensus to accept the paper. The reviewers concur that the authors addressed their primary concerns, making key clarifications. The AC sees no reason to override the consensus of the reviewers and is satisfied by the technical merit of the paper and its potential to make an impact on the field.

In addition to the revision items raised by the authors in their final response, I would request that they revise their claim that "the key problem we solve is optimal counterfactual probe generation". I concur with reviewer 5xm4 that a reasonable reader would expect this to be a mathematical claim of (global) optimality, which the authors clarify is not the case. Alternatives such as "high-quality" or similar would be more appropriate.